# Recursive Nested Filtering for Efficient Amortized Bayesian Experimental Design

**Sahel Iqbal**[*]
Aalto University

**Hany Abdulsamad**
Aalto University

**Sara Pérez-Vieites**
Aalto University

**Simo Särkkä**
Aalto University

**Adrien Corenflos**
University of Warwick

## Abstract

This paper introduces the Inside–Out Nested Particle Filter (IO–NPF), a novel fully recursive algorithm for amortized sequential Bayesian experimental design in the non-exchangeable setting. We frame policy optimization as maximum likelihood estimation in a non-Markovian state-space model, achieving (at most) $\mathcal{O}(T^2)$ computational complexity in the number of experiments. We provide theoretical convergence guarantees and introduce a backward sampling algorithm to reduce trajectory degeneracy. The IO–NPF offers a practical, extensible, and provably consistent approach to sequential Bayesian experimental design, demonstrating improved efficiency over existing methods.

## 1 Introduction

Bayesian experimental design (BED, Chaloner and Verdinelli, 1995; Lindley, 1956) is a mathematical framework for designing experiments to maximize information about the parameters of a statistical model. The primary objective of BED is to select a user-controllable input, known as the *design*, that is optimal in the sense that the resulting observation is, on average, maximally informative. This optimal design is obtained by solving a joint maximization and integration problem, which is typically intractable and thus requires approximation (Kueck et al., 2009). The difficulty is further compounded in sequential experiments, where the conventional *myopic* strategy of optimizing one experiment at a time is both suboptimal and time-consuming (Huan, Jagalur, et al., 2024; Rainforth et al., 2024).

An alternative to the myopic approach for sequential BED is to learn a *policy* instead of optimizing for individual designs (Foster et al., 2021; Huan and Marzouk, 2016; Ivanova et al., 2021). In this approach, we pay an upfront cost to learn the policy, but experiments can be performed in real-time. A recent addition to this literature is Iqbal et al. (2024), which leverages the duality between control and inference (Attias, 2003; Toussaint and Storkey, 2006) to rephrase policy optimization for BED as a maximum likelihood estimation (MLE) problem in a nonlinear, non-Markovian state-space model (SSM). Iqbal et al. (2024) also introduced the *Inside–Out (IO) SMC*[2] algorithm, a nested particle filter targeting this non-standard SSM.

The IO–SMC[2] was shown to be a sample-efficient alternative to state-of-the-art BED amortization methods (Iqbal et al., 2024). However, this algorithm has two key limitations. First, it is not a fully recursive algorithm, as it requires reprocessing all past observations for each experiment within an inner particle filter, which incorporates a Markov chain Monte Carlo (MCMC) kernel. Second, the transition density of this MCMC kernel cannot be computed, and generating samples from

---

[*]Correspondence address: sahel.iqbal@aalto.fi

Workshop on Bayesian Decision-making and Uncertainty, 38th Conference on Neural Information Processing Systems (NeurIPS 2024).

the smoothing distribution is done via genealogy tracking (Kitagawa, 1996, Section 4.1), which degenerates for long experiment sequences $T \gg 1$. In view of this, our contributions are as follows:

1. We propose a novel, *fully recursive*, provably consistent, nested particle filter that performs sequential BED, which we refer to as the Inside–Out NPF (IO–NPF). The IO–NPF replaces the MCMC kernel in IO–SMC$^2$ with the jittering kernel from the *nested particle filter* (NPF) of Crisan and Míguez (2017, 2018).

2. We propose a Rao-Blackwellized (Appendix A.2) backward sampling scheme (Godsill et al., 2004) for the IO–NPF to counter the degeneracy of the genealogy tracking smoother, thus improving the overall amortization performance.

## 2 Problem Setup

Let $\theta$ denote the parameters of interest, $\xi_t$ the design, and $x_t$ the outcome of the $t$-th experiment, where $t \in \{0, 1, \ldots, T\}$. For brevity, we define the augmented state $z_t := \{x_t, \xi_{t-1}\}$ for $t \geq 1$ and $z_0 := \{x_0\}$. We are interested in the non-exchangeable data setting where the outcomes, conditionally on $\theta$ and the designs $\xi_{0:T-1}$, follow Markovian dynamics,

$$p(x_{0:T} \mid \xi_{0:T-1}, \theta) = p(x_0) \prod_{t=1}^{T} f(x_t \mid x_{t-1}, \xi_{t-1}, \theta).$$

The designs $\xi_t$ are sampled from a stochastic policy $\pi_\phi(\xi_t \mid z_{0:t})$ that depends on the history until experiment $t$ and is parameterized by $\phi$. The joint density of the states, designs, and parameters can then be expressed as $p_\phi(z_{0:T}, \theta) = p(\theta)p(z_0) \prod_{t=1}^{T} p_\phi(z_t \mid z_{0:t-1}, \theta)$, where $p(z_0) = p(x_0)$ and $p_\phi(z_t \mid z_{0:t-1}, \theta) = f(x_t \mid x_{t-1}, \xi_{t-1}, \theta) \pi_\phi(\xi_{t-1} \mid z_{0:t-1})$.

Our goal is to optimize the *expected information gain* (EIG, Lindley, 1956) objective with respect to the policy parameters $\phi$. The EIG is defined as

$$\mathcal{I}(\phi) := \mathbb{E}_{p_\phi(z_{0:T})}\big[\mathbb{H}\big[p(\theta)\big] - \mathbb{H}[p(\theta \mid z_{0:T})]\big],$$

where $\mathbb{H}$ denotes the entropy of a distribution. Hence, the EIG represents the expected reduction in entropy in the parameters $\theta$ under the policy $\pi_\phi$, and the corresponding optimization problem is $\phi^* := \arg\max_\phi \mathcal{I}(\phi)$.

### 2.1 The Dual Inference Problem

If the noise in the dynamics is static, then, equivalently (Iqbal et al., 2024, Proposition 1),

$$\mathcal{I}(\phi) \equiv \mathbb{E}_{p_\phi(z_{0:T})}\left[\sum_{t=1}^{T} r_t(z_{0:t})\right], \tag{1}$$

where '$\equiv$' denotes equality up to an additive constant and

$$r_t(z_{0:t}) := -\log \int p(\theta \mid z_{0:t-1}) f(x_t \mid x_{t-1}, \xi_{t-1}, \theta) \, d\theta.$$

Following Toussaint and Storkey (2006), we define a *potential function* $g_t(z_{0:t}) := \exp\big\{\eta \, r_t(z_{0:t})\big\}$, where $\eta > 0$, and construct a non-Markovian Feynman-Kac model (Del Moral, 2004) $\Gamma_t$,

$$\Gamma_t(z_{0:t}; \phi) = \frac{1}{Z_t(\phi)} p(z_0) \prod_{s=1}^{t} p_\phi(z_s \mid z_{0:s-1}) \, g_s(z_{0:s}), \qquad 0 \leq t \leq T. \tag{2}$$

Here, $p_\phi(z_t \mid z_{0:t-1}) = \int p_\phi(z_t \mid z_{0:t-1}, \theta) \, p(\theta \mid z_{0:t-1}) \, d\theta$ are the marginal dynamics under the filtered posterior $p(\theta \mid z_{0:t-1})$ and $Z_t(\phi) = \int g_{1:t}(z_{0:t}) \, p_\phi(z_{0:t}) \, dz_{0:t}$ is the normalizing constant, with $g_{1:t}(z_{0:t}) = \prod_{s=1}^{t} g_s(z_{0:s})$.

Iqbal et al. (2024) showed that maximizing $Z_t(\phi)$ is equivalent to maximizing a proxy EIG objective. The authors performed this maximization by evaluating the score $\mathcal{S}(\phi)$ under samples from $\Gamma_T(\cdot; \phi)$

$$\mathcal{S}(\phi) := \nabla_\phi \log Z_T(\phi) = \int \nabla_\phi \log p_\phi(z_{0:T}) \, \Gamma_T(z_{0:T}; \phi) \, dz_{0:T},$$

and using a Markovian score climbing procedure (Naesseth et al., 2020) to perform policy optimization (see details in Appendix C). In Iqbal et al. (2024), sampling from $\Gamma_T(\cdot; \phi)$ was done by constructing a *non-recursive* particle filter, named Inside–Out SMC$^2$. The following section outlines a novel and efficient procedure to generate these samples in a *fully recursive* manner.

## 3 The Inside–Out Nested Particle Filter (IO–NPF)

For a bootstrap particle filter (Chopin and Papaspiliopoulos, 2020, Chapter 10) targeting $\Gamma_t$, we need to (a) draw importance samples from $p_\phi(z_t \mid z_{0:t-1})$, and (b) assign importance weights proportional to the potential function $g_t$ to the samples. Both these operations require computing integrals with respect to $p(\theta \mid z_{0:t-1})$, which is intractable in general. IO–SMC$^2$ provided a solution by nesting two particle filters: one to approximate $\Gamma_t$ and generating trajectories $\{z_{0:t}^n\}_{1 \le n \le N}$ for $t = 0, \ldots, T$; and another inner particle filter to approximate $p(\theta \mid z_{0:t}^n)$ with an empirical distribution,

$$p(\theta \mid z_{0:t}^n) \approx \hat{p}(\theta \mid z_{0:t}^n) := \frac{1}{M} \sum_{m=1}^{M} \delta_{\theta_t^{nm}}(\theta).$$

Here, $\delta_\theta$ is the Dirac delta function centered at $\theta$ and $\{\theta_t^{nm}\}_{1 \le m \le M}$ is the set of particles associated with the trajectory $z_{0:t}^n$. The marginal transition density $p_\phi$ and the potential $g_t$ are then approximated by computing the necessary integrals with respect to this approximate filtering distribution $\hat{p}(\theta \mid z_{0:t}^n)$. This approximate distribution is obtained via a particle filter using a Markov kernel $\kappa_t$ targeting $p(\theta \mid z_{0:t}^n)$. This means that the sequence of observations and designs has to be reprocessed from scratch, leading to a computational complexity of $\mathcal{O}(t)$ at every time step.

An alternative, which is used in the nested particle filter (NPF) of Crisan and Míguez (2018) to achieve a recursive algorithm, and which we borrow for the IO–NPF, is to jitter (Liu and West, 2001) the $\theta$-particles using a Markov kernel $\kappa_M$ that has $\mathcal{O}(1)$ cost. The jittered particles will no longer be samples from the true posterior, but by scaling the perturbation induced by the kernel as a function of the number of samples $M$, we can control the error incurred through jittering (see Crisan and Míguez, 2018, Section 5.1).

The Inside–Out NPF algorithm, presented in Algorithm 1 (Appendix A), is a nested particle filter targeting $\Gamma_t$ with the same algorithmic structure as IO–SMC$^2$, but with the jittering kernel from the NPF used to rejuvenate the $\theta$–particles. This seemingly small algorithmic change leads to two major consequences. First, it makes the algorithm fully recursive, with $\mathcal{O}(NMT)$ computational complexity against the $\mathcal{O}(NMT^2)$ of IO–SMC$^2$. Second, it allows us to construct a backward sampling algorithm to generate less degenerate trajectories from $\Gamma_t$, which we describe in Section 3.1.

We adapt results from Crisan and Míguez (2018) to establish the following proposition, proved in Appendix B, for the consistency of the IO–NPF.

**Proposition 1** (Consistency). *Let $\Gamma_t^M(z_{0:t})$ denote the marginal target distribution of Algorithm 1. Under technical conditions listed in Appendix B, for all bounded functions $h$, we have*

$$\lim_{M \to \infty} \mathbb{E}_{\Gamma_t^M}\left[h(z_{0:t})\right] = \mathbb{E}_{\Gamma_t}\left[h(z_{0:t})\right].$$

### 3.1 Backward Sampling

The trajectories $z_{0:T}^n$ generated by tracing particle genealogies are valid samples from $\Gamma_T$. However, for $T \gg 1$, very few unique paths for the first components of the trajectories $z_{0:T}^n$ may remain, a problem due to resampling known as path degeneracy. The backward sampling algorithm (Godsill et al., 2004) is one way to circumvent the path degeneracy problem by simulating smoothing trajectories backward in time.

Before we explain the algorithm, we first note that at time $t$, one "particle" of the IO–NPF is the set of random variables $y_t^n := \left\{ z_t^n, \{\theta_t^{nm}, B_t^{nm}\}_{1 \le m \le M} \right\}$, where $B_t^{nm}$ is the ancestor index for $\theta_t^{nm}$. Given a partial backward trajectory $\bar{y}_{t+1:T}$ which we have already simulated and partial forward trajectories $(y_{0:t}^n)_{n=1}^N$, backward sampling selects $\bar{y}_t$ from $(y_t^n)_{n=1}^N$ with probability proportional to

$$W(y_{0:t}^n, \bar{y}_{t+1:T}) W_{z,t}^n := \frac{\Gamma_T^M(y_{0:t}^n, \bar{y}_{t+1:T})}{\Gamma_t^M(y_{0:t}^n)} W_{z,t}^n, \qquad \text{for } t \ge 0,$$

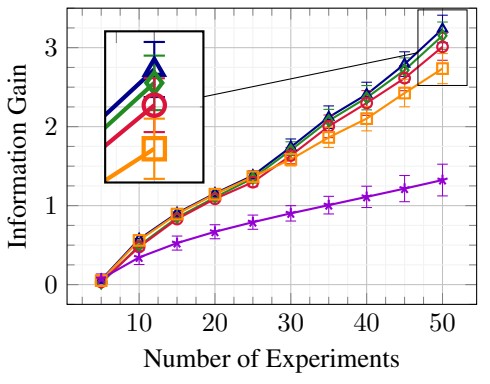

Figure 1: Accumulation of the *realized* information gain computed in closed form for different policies on the conditionally linear stochastic pendulum with a Gaussian parameter prior. We report the mean and standard deviation over 1024 realizations. Here IO–SMC$^2$ (Exact) —▲— represents an ideal baseline where $\theta$–posterior updates are closed-form, which is only possible in conjugate settings. Otherwise, our IO–NPF with backward sampling (BS) —◆— outperforms all alternatives.

| —▲— IO–SMC$^2$ (Exact) | —◆— IO–NPF + BS | —○— IO–SMC$^2$ | —□— IO–NPF | —✴— Random |

where $\Gamma_T^M$ is the true target distribution of the IO–NPF (see Appendix A.1) and $W_{z,t}^n$ are the filtering weights. This ratio can be computed by considering the transition densities of its constituent steps in Algorithm 1. In practice, implementing this naively would incur two drawbacks.

First, while correct, using $W(y_{0:t}^n, \bar{y}_{t+1:T})$ *as is* would result in a degenerate algorithm: this is because the $\theta$–particles $\{\theta_t^{nm}, \theta_{t+1}^{nm}\}$ are unlikely to be compatible pairwise, even when the populations $\{(\theta_t^{nm})_{m=1}^M, (\theta_{t+1}^{nm})_{m=1}^M\}$ are. In Appendix A.2 we show how this can be solved by marginalizing the weight over the ancestry of the $\theta$–particles.

Second, the overall cost of backward sampling would be $\mathcal{O}(T^2 N^2 M^2)$. This is prohibitive, and we prefer a cheaper 'sparse' $\mathcal{O}(N)$ alternative consisting of sampling from a Markov kernel (again!) that keeps the distribution of the ancestors invariant (Bunch and Godsill, 2013; Dau and Chopin, 2023). This procedure is detailed in Appendix A.2.

## 4 Numerical Validation

To validate our method numerically, we consider the stochastic dynamics of a pendulum in conditionally linear-Gaussian form. The unknown parameters are a combination of its mass and length, while the measurement of the angular position and velocity constitutes the outcome of an experiment. The design is the torque applied to the pendulum at every experiment. We consider a series of 50 experiments, corresponding to 50 time steps of pendulum dynamics. The prior on the parameters is Gaussian.

Table 1: EIG, sPCE and runtime for various BED schemes. Mean and standard deviation over 25 seeds.

| Policy | EIG Estimate (1) | sPCE | Runtime [$s$] |
|---|---|---|---|
| Random | $1.37 \pm 0.08$ | $1.44 \pm 0.35$ | – |
| iDAD | $2.58 \pm 0.17$ | $2.53 \pm 0.35$ | – |
| IO–NPF | $2.98 \pm 0.19$ | $3.12 \pm 0.35$ | $0.34 \pm 0.01$ |
| IO–SMC$^2$ | $3.35 \pm 0.20$ | $3.44 \pm 0.39$ | $5.71 \pm 0.08$ |
| IO–NPF + BS | $3.47 \pm 0.18$ | $3.54 \pm 0.40$ | $2.73 \pm 0.07$ |
| IO–SMC$^2$ Exact | $3.54 \pm 0.20$ | $3.64 \pm 0.38$ | $1.36 \pm 0.04$ |

Table 1 compares the *expected* information gain and its surrogate, the *sequential prior contrastive estimate* (sPCE, Foster et al., 2021), for a variety of amortized BED algorithms. The EIG estimate is computed by leveraging our nested sampling scheme to draw samples from the marginal $p_\phi(z_{0:T})$ and evaluate objective (1). The comparison includes policies trained using the IO–NPF both with and without backward sampling (BS), IO–SMC$^2$, implicit deep adaptive design (iDAD, Ivanova et al., 2021), and a random policy baseline. Furthermore, Figure 1 shows the *realized* information gain achieved by policies trained using different algorithms over the number of experiments. In this particular conjugate setting, the $\theta$–posteriors and the marginal transition densities can be computed in closed form, making the exact variant of IO–SMC$^2$ — which substitutes the inner particle filter with exact posterior computation — an ideal baseline. The results, both in Table 1 and Figure 1, highlight the benefit of backward sampling: IO–NPF *without* backward sampling performs worse than IO–SMC$^2$, whereas IO–NPF *with* backward sampling outperforms it.

Finally, Table 1 presents the runtime statistics of a single amortization iteration of the various inside–out particle filtering algorithms, emphasizing IO–NPF with backward sampling as the most

favorable option that offers the best balance between task performance, computational efficiency, and generality. For more details on this evaluation consult Appendix D. A public implementation in the Julia Programming Language is available at `https://github.com/Sahel13/InsideOutNPF.jl`.

## 5 Outlook and future work

We strongly believe that this new, fully recursive perspective on amortized Bayesian experimental design offers a promising, practical, extensible, and well-behaved approach both statistically and empirically. The main limitation of our approach (and of IO–SMC$^2$) is the need to know the Markovian outcome-likelihood, which may not always be available. We hope to address this in future research. Additionally, we aim to derive stronger, non-asymptotic bounds for the error of the IO–NPF, similar to Míguez et al. (2013), in particular by incorporating recent results on backward sampling stability (Dau and Chopin, 2023), providing stronger guarantees on the learned policies.

## 6 Individual Contributions

AC and HA initially conceived the idea for this article, with AC and SI then developing the methodology. SI proved the consistency of the method, following AC's suggestions. The code implementation and experiments were carried out jointly by HA and SI. SI took the lead in writing the article, with valuable help from AC and HA. SPV and SS provided helpful discussions and feedback on the manuscript. All authors reviewed and validated the final version of the manuscript.

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

## A  Details of the Inside–Out NPF

---

**Algorithm 1:** The Inside–Out NPF algorithm.

---

**notation:** Operations involving the superscript $n$ are to be performed for $n = 1, \ldots, N$.

1  Sample $z_0^n \sim p(z_0)$, and set $W_{z,0}^n \leftarrow 1/N$.

2  Sample $\theta_0^{nm} \sim p(\theta)$ and set $W_{\theta,0}^{nm} \leftarrow 1/M$ for $m = 1, \ldots, M$.

3  **for** $t \leftarrow 1$ **to** $T$ **do**

4  $\quad$ Sample $A_t^n \sim \mathcal{M}(W_{z,t-1}^{1:N})$. $\hfill$ // Resample

5  $\quad$ Sample $z_t^n \sim p_\phi^M(z_t \mid z_{0:t-1}^{A_t^n})$. $\hfill$ // Propagate

6  $\quad$ Set $W_{z,t}^n \propto g_t^M(z_{0:t}^n)$. $\hfill$ // Reweight

7  $\quad$ **for** $m \leftarrow 1$ **to** $M$ **do**

8  $\quad\quad$ Set $W_{\theta,t}^{nm} \propto p(z_t^n \mid z_{0:t-1}^{A_t^n}, \theta_{t-1}^{A_t^n m})$. $\hfill$ // Reweight

9  $\quad\quad$ Sample $B_t^{nm} \sim \mathcal{M}(W_{\theta,t}^{n 1:M})$. $\hfill$ // Resample

10 $\quad\quad$ Sample $\theta_t^{nm} \sim \kappa_M(\theta_t \mid \theta_{t-1}^{A_t^n B_t^{nm}})$. $\hfill$ // Jitter

11 $\quad$ **end**

12 **end**

---

The Inside–Out NPF is presented in Algorithm 1. It is a nested particle filter to sample from the Feynman-Kac model $(\Gamma_t)_{t=0}^T$ given by (2). At a time step $t$, the algorithm provides weighted trajectories $(W_{z,t}^n, z_{0:t}^n)_{n=1}^N$, with each trajectory computed recursively as $z_{0:t}^n := (z_{0:t-1}^{A_t^n}, z_t^n)$. Furthermore, for every $n \in \{1, \ldots, N\}$, it generates a particle approximation of $p(\theta \mid z_{0:t}^n)$,

$$p(\theta \mid z_{0:t}^n) \approx \hat{p}(\theta \mid z_{0:t}^n) := \frac{1}{M} \sum_{m=1}^M \delta_{\theta_t^{nm}}(\theta).$$

The marginal transition density for the state $x_t$ is then approximated as

$$p^M(x_{t+1} \mid z_{0:t}^n, \xi_t) := \int f(x_{t+1} \mid x_t^n, \xi_t, \theta) \, \hat{p}(\theta \mid z_{0:t}^n) \, \mathrm{d}\theta = \frac{1}{M} \sum_{m=1}^M f(x_{t+1} \mid x_t^n, \xi_t, \theta_t^{mn}).$$

We then define

$$p_\phi^M(z_{t+1} \mid z_{0:t}^n, \xi_t) := p^M(x_{t+1} \mid z_{0:t}^n, \xi_t) \, \pi_\phi(\xi_t \mid z_{0:t}^n),$$

$$g_t^M(z_{0:t+1}^n) := \exp\left\{ -\eta \log p^M(x_{t+1}^n \mid z_{0:t}^{A_t^n}, \xi_t) \right\}.$$

These approximations are used for the importance sampling and weighting steps (lines 5 and 6) in Algorithm 1. Finally, $\mathcal{M}(W^{1:N})$ refers to the multinomial distribution over the integer set $\{1, \ldots, N\}$ with corresponding weights $\{W^1, \ldots, W^N\}$.

### A.1  Target distribution of the IO–NPF

We follow a similar analysis to that in Iqbal et al. (2024, Section 4.3). We drop the $n$ indices and the explicit dependence on $\phi$. A "particle" of the IO–NPF at time $t$ is the random variable $(z_t, \theta_t^{1:M}, B_t^{1:M})$, with $B_0^{1:M} := \Phi$ being the null set. At $t = 0$, they have a density

$$\Gamma_0^M(z_0, \theta_0^{1:M}) = p(z_0) \prod_{m=1}^M p(\theta_0^m).$$

For $t \geq 1$, the ratio of successive target densities can be broken down as

$$\frac{\Gamma_{t+1}^M(z_{0:t+1}, \theta_{0:t+1}^{1:M}, B_{1:t+1}^{1:M})}{\Gamma_t^M(z_{0:t}, \theta_{0:t}^{1:M}, B_{1:t}^{1:M})} = \frac{\Gamma_{t+1}^M(z_{0:t+1}, \theta_{0:t}^{1:M}, B_{1:t}^{1:M})}{\Gamma_t^M(z_{0:t}, \theta_{0:t}^{1:M}, B_{1:t}^{1:M})} \times \frac{\Gamma_{t+1}^M(z_{0:t+1}, \theta_{0:t+1}^{1:M}, B_{1:t+1}^{1:M})}{\Gamma_{t+1}^M(z_{0:t+1}, \theta_{0:t}^{1:M}, B_{1:t}^{1:M})}. \quad (3)$$

The second fraction in (3) corresponds to the resample and jitter steps in the inner particle filter:

$$\frac{\Gamma_{t+1}^M(z_{0:t+1}, \theta_{0:t+1}^{1:M}, B_{1:t+1}^{1:M})}{\Gamma_{t+1}^M(z_{0:t+1}, \theta_{0:t}^{1:M}, B_{1:t}^{1:M})} = \prod_{m=1}^M W_{\theta,t+1}^{B_{t+1}^m} \kappa_M(\theta_{t+1}^m \mid \theta_t^{B_{t+1}^m}), \quad (4)$$

---

**Algorithm 2:** MCMC backward sampler with independent Metropolis-Hastings (Bunch and Godsill, 2013).

---

**input:** All random variables generated by Algorithm 1.
**output:** A trajectory $\bar{y}_{0:T}$.
**notation:** Operations involving the superscripts $n$ are to be performed for $n = 1, \ldots, N$.

**1** Sample $I_T \sim \mathcal{M}(W_{z,T}^{1:N})$.
**2** Set $\bar{y}_T \leftarrow (z_T^{I_T}, \theta_T^{I_T 1:M})$.
**3 for** $t \leftarrow T - 1$ **to** 0 **do**
**4** $\quad$ Sample $I_t \sim \mathcal{M}(W_{z,t}^{1:N})$.
**5** $\quad$ Accept $I_t$ with probability $\min \left\{ 1, W\left( y_{0:t}^{I_t}, \bar{y}_{t+1:T} \right) / W\left( y_{0:t}^{A_{t+1}^{I_{t+1}}}, \bar{y}_{t+1:T} \right) \right\}$.
**6** $\quad$ If rejected, set $I_t \leftarrow A_{t+1}^{I_{t+1}}$. $\qquad$ // i.e., fall back to ancestor tracing
**7** $\quad$ Set $\bar{y}_t \leftarrow (z_t^{I_t}, \theta_t^{I_t 1:M})$.
**8 end**

---

while the first fraction in (3) corresponds to the propagate and reweight steps of the outer particle filter,

$$
\frac{\Gamma_{t+1}^M(z_{0:t+1}, \theta_{0:t}^{1:M}, B_{1:t}^{1:M})}{\Gamma_t^M(z_{0:t}, \theta_{0:t}^{1:M}, B_{1:t}^{1:M})} \propto \left\{ \frac{1}{M} \sum_{m=1}^M p(z_{t+1} \mid z_{0:t}, \theta_t^m) \right\} \tag{5}
$$
$$
\times \exp \left\{ -\eta \log \left( \frac{1}{M} \sum_{m=1}^M f(x_{t+1} \mid x_t, \xi_t, \theta_t^m) \right) \right\}.
$$

### A.2  Backward sampling

In order to reduce degeneracy of particle smoothing (and conditional SMC), a common strategy is to employ backward or ancestor sampling (Lindsten et al., 2014; Whiteley, 2010). This typically means that, instead of tracing back ancestors as is done in Iqbal et al., 2024, we can ask the question "what ancestor could have resulted in this particle?" and simulate accordingly. This is often done using direct simulation as per Godsill et al. (2004), costing $\mathcal{O}(N)$.

An issue with this is that its cost becomes prohibitive as soon as

1. we wish to compute expectations over the CSMC chain, whereby we may want to obtain more than a single trajectory at each iteration, and the cost is $\mathcal{O}(N)$ *per desired sample*,

2. the cost of evaluating the probability of a particle having a given ancestor is high.

For these two reasons, the first one coming from the Rao–Blackwellization already proposed in Iqbal et al. (2024) but with ancestor tracing, the second due to reasons explained later in this section, we prefer a 'sparse' version of backward sampling, where we do not need to evaluate all possible combinations, which we now present.

In this section, we describe the MCMC backward sampler from Bunch and Godsill (2013) and Dau and Chopin (2023), presented in Algorithm 2, and how we adapt it for the IO–NPF. For notational convenience, let us define $y_t^n \coloneqq \left( z_t^n, \theta_t^{n 1:M}, B_t^{n 1:M} \right)$. As mentioned in the previous section, this $y_t^n$ denotes one "particle" of our nested particle filter. At time $t$, we will have already sampled backward indices $I_{t+1:T}$. Denoting $\bar{y}_t = y_t^{I_t}$ for all $t \in \{0, \ldots, T\}$, we start from the true ancestor of $\bar{y}_{t+1}$, $I_t = A_{t+1}^{I_{t+1}}$, and apply an independent Metropolis-Hastings step targeting the invariant density

$$
\Gamma_T^M (i \mid y_0^{1:N}, \ldots, y_T^{1:N}, A_1^{1:N}, \ldots, A_T^{1:N}, I_{t+1:T}) \propto W_{z,t}^i W(y_{0:t}^i, \bar{y}_{t+1:T}),
$$

where $W_{z,t}^{1:N}$ are the filtering weights (line 6 in Algorithm 1), and $W(y_{0:t}^n, \bar{y}_{t+1:T})$ is defined as

$$
\begin{aligned}
W(y_{0:t}^n, \bar{y}_{t+1:T}) &:= \frac{\Gamma_T^M(y_{0:t}^n, \bar{y}_{t+1:T})}{\Gamma_t^M(y_{0:t}^n)} \\
&= \prod_{s=t+1}^T \Gamma_s^M(\bar{y}_s \mid y_{0:t}^n, \bar{y}_{t+1:s-1}) \\
&= \prod_{s=t+1}^T \Gamma_s^M(\bar{z}_s \mid y_{0:t}^n, \bar{y}_{t+1:s-1}) \, \Gamma_s^M(\bar{\theta}_s^{1:M}, \bar{B}_s^{1:M} \mid \bar{z}_s, y_{0:t}^n, \bar{y}_{t+1:s-1}). \quad (6)
\end{aligned}
$$

The first term is the ratio in (5), corresponding to the importance sampling and weighting steps in the outer particle filter. The second term in (6) is the transition density of the particles in the inner particle filter from (4). When using this cheaper formulation of backward sampling, we can obtain $N$ trajectories at the fixed cost of $\mathcal{O}(N)$, rather than $\mathcal{O}(N^2)$ as given by the direct approach of Godsill et al. (2004).

While computing $W(y_{0:t}^n)$ following (6) would result in a valid algorithm, it would likely be degenerate. Indeed, the second term of the product, given by (4) would result in very low transition probabilities. This is because, even if the particles may represent the same posterior distribution, their pairwise alignment may be arbitrarily poor. Thankfully, it is possible to integrate the distribution over the weights $B_t^{1:M}$, in effect Rao–Blackwellizing the weights $W(y_{0:t}^n)$, which we describe next.

Let us omit the $n$ superscripts and consider the transition probability of the $\theta$ particles,

$$
\Gamma_{t+1}^M(\theta_{t+1}^{1:M} \mid B_{t+1}^{1:M}, \theta_t^{1:M}) = \prod_{m=1}^M \kappa_M(\theta_{t+1}^m \mid \theta_t^{B_{t+1}^m}).
$$

The probabilities above are also conditioned on a specific trajectory $z_{0:t+1}$ and all $\theta$ and $B$ particles from the previous time steps, but we omit these for notational clarity. We now marginalize over the resampling indices $B_{t+1}^{1:M}$ to get

$$
\begin{aligned}
\Gamma_{t+1}^M(\theta_{t+1}^{1:M} \mid \theta_t^{1:M}) &= \int \prod_{m=1}^M \kappa_M(\theta_{t+1}^m \mid \theta_t^{B_{t+1}^m}) \, \mathrm{d}\Gamma_{t+1}^M(B_{t+1}^{1:M}) \\
&= \prod_{m=1}^M \int \kappa_M(\theta_{t+1}^m \mid \theta_t^{B_{t+1}^m}) \, \mathrm{d}\Gamma_{t+1}^M(B_{t+1}^{1:M}) \\
&= \prod_{m=1}^M \left\{ \sum_{k=1}^M W_{\theta,t+1}^k \, \kappa_M(\theta_{t+1}^m \mid \theta_t^k) \right\}.
\end{aligned}
$$

In the second line, we have used the fact when using multinomial resampling, the resampling indices are independent and identically distributed.

*Remark.* The use of the MCMC backward sampler of (Bunch and Godsill, 2013) within CSMC is non-standard, and to the best of our knowledge, it has only been studied within the context of particle smoothing only (Dau and Chopin, 2023). This is because, in general, it is not very useful there: the cost of sampling once from (Bunch and Godsill, 2013) is the same as sampling $N$ times: $\mathcal{O}(N)$, which is the same as sampling a single trajectory using standard backward sampling (Godsill et al., 2004). In particular, it may not directly seem obvious to the reader why the algorithm is correct *at all*. This is due to the fact that the backward sampling step in conditional SMC can be seen as the second step of a form of 'Hastings-within-Gibbs' algorithm, where the first, Hastings, step is given by standard the forward pass of CSMC, and the second, Gibbs, step is an exact sampling from the resulting categorical distribution $\mathrm{Cat}\left(W(y_{0:t}^n, \bar{y}_{t+1:T})\right)$ of the ancestors (Whiteley, 2010). This second part can also be implemented using a Markov kernel keeping the distribution of the ancestors invariant, which is what we do here.

## B  Proof of Proposition 1

*Notation and setup.* We assume that $\theta \in D_\theta$, where $\mathbb{R}^d \supset D_\theta$ is a compact set. $B(\mathbb{R}^d)$ denotes the set of bounded, measurable functions defined on $\mathbb{R}^d$. If $f$ is a $\nu$-integrable function, $\nu f := \int f \mathrm{d}\nu$.

---

**Algorithm 3:** A particle filter for static models.

---

**notation:** Operations involving the superscript $m$ are to be performed for $m = 1, \ldots, M$.

**input:** A trajectory $z_{0:T}$.

1   $\theta_0^m \sim p(\theta)$.

2 **for** $t \leftarrow 1$ **to** $T$ **do**

3      $W_t^m \propto p(z_t \mid z_{0:t-1}, \theta_{t-1}^m)$.

4      $B_t^m \sim \mathcal{M}(W_t^{1:M})$.

5      $\theta_t^m \sim \kappa_M(\mathrm{d}\theta \mid \theta_{t-1}^{B_t^m})$.

6 **end**

---

For a function $h : S \to \mathbb{R}$, the supremum norm is denoted as $\|h\|_\infty := \sup_{x \in S} |h(x)|$. The $L^p$ norm of a random variable $Z$ defined on the probability space $(\Omega, \mathcal{F}, \nu)$ is given by

$$\|Z\|_p := \mathbb{E}\big[|Z|^p\big]^{1/p} = \left[\int |Z|^p \, \mathrm{d}\nu\right]^{1/p}.$$

We use $\xrightarrow{\text{a.s.}}$ to denote convergence almost surely and $\xrightarrow{\mathbb{P}}$ for convergence in probability.

To prove Proposition 1, we first bound the error in approximating the filtering distribution of $\theta$ with a particle filter that uses the jittering kernel. Consider the particle filter given in Algorithm 3, which corresponds to the inner particle filter in the IO–NPF (Algorithm 1). For a fixed trajectory $z_{0:T}$, at each time step $t$ it yields an empirical measure

$$\mu_t^M(\mathrm{d}\theta) = \frac{1}{M} \sum_{m=1}^{M} \delta_{\theta_t^m}(\mathrm{d}\theta)$$

that approximates the filtering distribution $\mu_t(\mathrm{d}\theta) := p(\theta \mid z_{0:t}) \, \mathrm{d}\theta$. To bound the error in estimating integrals using $\mu_t^M$, we make the following assumptions.

**Assumption 1.** The kernel $\kappa_M(\mathrm{d}\theta \mid \theta'), \theta' \in D_\theta$ satisfies the inequality

$$\sup_{\theta' \in D_\theta} \int |h(\theta) - h(\theta')| \, \kappa_M(\mathrm{d}\theta \mid \theta') \leq \frac{c_\kappa \|h\|_\infty}{\sqrt{M}}$$

for any $h \in B(D_\theta)$ and some constant $c_\kappa < \infty$.

This assumption bounds the expected absolute difference of integrable functions with respect to the jittering kernel, and is one of the key assumptions made in Crisan and Míguez (2018) for the convergence analysis of the nested particle filter (NPF).

**Assumption 2.** For all $z_{0:t}$, $t \geq 1$ and $\theta \in D_\theta$, the function $\theta \mapsto p(z_t \mid z_{0:t-1}, \theta)$ is positive and bounded.

We now bound the approximation errors in $L^p$ using the following proposition.

**Proposition 2.** *Under Assumptions 1 and 2, for all $h \in B(D_\theta)$ and $1 \leq p < \infty$, we have*

$$\|\mu_t^M h - \mu_t h\|_p \leq \frac{c_t \|h\|_\infty}{\sqrt{M}}, \tag{7}$$

*where $c_t < \infty$ is a constant independent of $M$.*

*Proof.* We prove this via induction. Since the particles $\theta_0^m \sim \mu_0$ are sampled i.i.d, the result holds for $t = 0$ using the Marcinkiewicz–Zygmund inequality. Let us assume it holds at time $t-1$ for some $t \geq 1$. We define the intermediate measure

$$\tilde{\mu}_t^M(\mathrm{d}\theta) := \frac{1}{M} \sum_{m=1}^{M} \delta_{\theta_{t-1}^{B_t^m}}(\mathrm{d}\theta),$$

representing the empirical measure after reweighting and resampling but before jittering. Using Minkowski's inequality, we have

$$\|\mu_t^M h - \mu_t h\|_p \leq \|\mu_t^M h - \tilde{\mu}_t^M h\|_p + \|\tilde{\mu}_t^M h - \mu_t h\|_p. \tag{8}$$

The first term in the RHS of (8) can be bounded using Lemma 3 from Crisan and Míguez (2018) (Eqs. 5.7 and 5.9), which relies on Assumption 1, to get

$$\|\mu_t^M h - \tilde{\mu}_t^M h\|_p \leq \frac{\tilde{c}_1 \|h\|_\infty}{\sqrt{M}},$$

where $\tilde{c}_1 < \infty$ is independent of $M$. For the second term in the RHS in (8), note that $\tilde{\mu}_t^M$ is obtained from $\mu_{t-1}^M$ through importance weighting and multinomial resampling. Under the assumption of bounded weight functions (Assumption 2), we have the following inequality from standard particle filter convergence results (see, for e.g., the proof of Lemma 1 in Míguez et al., 2013)

$$\|\tilde{\mu}_t^M h - \mu_t h\|_p \leq \frac{\tilde{c}_2 \|h\|_\infty}{\sqrt{M}},$$

where $\tilde{c}_2 < \infty$ is also independent of $M$. Putting it together, we get

$$\|\mu_t^M h - \mu_t h\|_p \leq \frac{c_t \|h\|_\infty}{\sqrt{M}},$$

with $c_t = \tilde{c}_1 + \tilde{c}_2$. □

**Corollary 1.** *Under the same assumptions as for Proposition 2, for all $h \in B(D_\theta)$,*

$$\mu_t^M h \xrightarrow{\text{a.s.}} \mu_t h.$$

*Proof.* We follow the proof technique of Crisan (2001), see also Chopin and Papaspiliopoulos (2020, Section 11.1). We start with Equation (7),

$$\|\mu_t^M h - \mu_t h\|_p \leq \frac{c_t \|h\|_\infty}{\sqrt{M}} =: \frac{C}{\sqrt{M}}, \quad \forall p \in [1, \infty).$$

Using Chebyshev's inequality, for all $\epsilon > 0$ and $1 \leq p < \infty$, we have

$$\mathbb{P}\big[|\mu_t^M h - \mu_t h| \geq \epsilon\big] \leq \frac{\mathbb{E}\big[|\mu_t^M h - \mu_t h|^p\big]}{\epsilon^p} \leq \frac{C^p}{M^{p/2}\epsilon^p}.$$

For $p > 2$, the following sum converges:

$$\sum_{M=1}^\infty \mathbb{P}\big[|\mu_t^M h - \mu_t h| \geq \epsilon\big] \leq \sum_{M=1}^\infty \frac{C^p}{M^{p/2}\epsilon^p} < \infty, \quad \forall p \in (2, \infty).$$

Thus, using the Borel–Cantelli lemma, the event $|\mu_t^M h - \mu_t h| \geq \epsilon$ occurs for only finitely many $M$, almost surely. Since this holds for any $\epsilon > 0$, we have the result

$$\mu_t^M h \xrightarrow{\text{a.s.}} \mu_t h.$$

□

Corollary 1 establishes that expectations under the empirical measures generated by the inner particle filter converge almost surely to the corresponding expectations under the true filtering distribution of $\theta$. We use this, in conjunction with Assumptions 2 and 3, to prove Proposition 1.

**Assumption 3.** For all $z_{0:t+1}, t \geq 1$ and $\theta \in D_\theta$, the function $\theta \mapsto f(x_{t+1} \mid x_t, \xi_t, \theta)$ is positive and bounded.

**Proposition 3** (Consistency)**.** *Let $\Gamma_t^M(z_{0:t})$ denote the marginal target distribution of Algorithm 1. Under Assumptions 2 and 3, for all $h \in B(\mathbb{R}^{d_x(t+1)})$, we have*

$$\lim_{M \to \infty} \Gamma_t^M h = \Gamma_t h.$$

*Proof.* The proof is identical to that of Iqbal et al. (2024, Proposition 2), since we have already proved weak convergence for the measures $\mu_t^M$ (Corollary 1). □

**Algorithm 4:** Markovian score climbing (Gu and Kong, 1998; Naesseth et al., 2020).

**input:** Initial trajectory $z_{0:T}^0$, initial parameters $\phi_0$, step sizes $\{\gamma_i\}_{i \in \mathbb{N}}$, Markov kernel $\mathcal{K}$.
**output:** Local optimum $\phi^*$ of the marginal likelihood.

1  Set $k \leftarrow 1$.
2  **while** *not converged* **do**
3      Sample $z_{0:T}^k \sim \mathcal{K}_{\phi_{k-1}}(\cdot \mid z_{0:T}^{k-1})$.
4      Compute $\hat{\mathcal{S}}(\phi_{k-1}) \leftarrow \nabla_\phi \log p_\phi(z_{0:T}^k)|_{\phi=\phi_{k-1}}$.
5      Update $\phi_k \leftarrow \phi_{k-1} + \gamma_k \hat{\mathcal{S}}(\phi_{k-1})$.
6      $k \leftarrow k + 1$.
7  **end**
8  **return** $\phi_k$

## C  Details on Policy Amortization

As mentioned in Section 2.1, our goal is to maximize the normalizing constant $Z(\phi)$ of the non-Markovian Feynman-Kac model $(\Gamma_t)_{t=0}^T$ as a proxy to the EIG objective (1)

$$\Gamma_t(z_{0:t}; \phi) = \frac{1}{Z_t(\phi)} p(z_0) \prod_{s=1}^t p_\phi(z_s \mid z_{0:s-1}) \, g_s(z_{0:s}), \qquad 0 \le t \le T. \tag{9}$$

We will refer to $Z(\phi)$ as the marginal likelihood. Let us define the *score function* $\mathcal{S}(\phi) := \nabla_\phi \log Z_T(\phi)$ to be the derivative of the log marginal likelihood. Using Fisher's identity (Cappé et al., 2005), we have

$$\mathcal{S}(\phi) = \nabla_\phi \log Z_T(\phi) = \int \nabla_\phi \log \tilde{\Gamma}_T(z_{0:T}; \phi) \, \Gamma_T(z_{0:T}; \phi) \, \mathrm{d}z_{0:T}$$

$$= \int \nabla_\phi \log p_\phi(z_{0:T}) \, \Gamma_T(z_{0:T}; \phi) \, \mathrm{d}z_{0:T},$$

where $\tilde{\Gamma}_T(z_{0:T}; \phi) := p_\phi(z_{0:T}) \, g_{1:T}(z_{0:T})$ is the un-normalized density from (9).

To maximize $Z(\phi)$, we use the Markovian score climbing (MSC) algorithm from Gu and Kong (1998) and Naesseth et al. (2020), presented in Algorithm 4. Given a Markov kernel $\mathcal{K}_\phi$ that is $\Gamma_T(\cdot; \phi)$-ergodic, MSC draws a sample from $\mathcal{K}_\phi$, evaluates the score under the sample, and updates the parameters $\phi$. This is repeated until convergence of the policy parameters. Algorithm 4 is guaranteed to converge to a local optimum of the marginal likelihood (Naesseth et al., 2020, Proposition 1).

We construct the Markov kernel $\mathcal{K}_\phi$ as a Rao-Blackwellized conditional sequential Monte Carlo kernel (Abdulsamad et al., 2023; Andrieu et al., 2010; Cardoso et al., 2023; Olsson and Westerborn, 2017). It is implemented as a simple modification to the IO–NPF algorithm, equivalent to the following operation. Given a *reference trajectory*, at each time step in the filter, we sample $N-1$ particles conditionally on the reference trajectory having survived the resampling step. For a thorough description of the algorithm, we refer to Iqbal et al. (2024, Appendix C.4).

## D  Details of Numerical Validation

### D.1  Stochastic pendulum environment

In this pendulum environment, the vector $x_t = [q_t, \dot{q}_t]^\top$ denotes the state of the pendulum, with $q_t$ being the angle from the vertical and $\dot{q}_t$ the angular velocity. The parameters of interest are $(m, l)$, the mass and length of the pendulum, while $g = 9.81$ and $d = 0.1$ are the gravitational acceleration and a damping constant. The design, $\xi_t \in [-1, 1]$, is the torque applied to the pendulum.

In order to compare with the exact variant of IO–SMC$^2$ (Iqbal et al., 2024), we transform the pendulum equations to obtain a conditional linear form, leading to the transformed parameter vector

$$\theta = \left[ \frac{3g}{2l}, \frac{3d}{ml^2}, \frac{3}{ml^2} \right]^\top.$$

The dynamics is described by the following Ito stochastic differential equation (SDE)

$$\mathrm{d}x_t = h(x_t, \xi_t)^\top \theta \, \mathrm{d}t + L \, \mathrm{d}\beta,$$

with a drift term $h(x_t, \xi_t) = [-\sin(q), -\dot{q}, \xi_t]^\top$, diffusion term $L = [0, 0.1]^\top$ and Brownian motion $\beta$. We descritze this SDE in time using the Euler-Maruyama scheme (Särkkä and Solin, 2019) with a step size $\mathrm{d}t = 0.05$ and consider a horizon of $T = 50$ experiments (time steps). The initial state is fixed to $x_0 = [0, 0]^\top$. Finally, to maintain conjugacy, we assume a Gaussian prior over $\theta$

$$p(\theta) = \mathrm{Normal}\left(\begin{bmatrix} 14.7 \\ 0 \\ 3.0 \end{bmatrix}, \begin{bmatrix} 0.1 & 0 & 0 \\ 0 & 0.01 & 0 \\ 0 & 0 & 0.1 \end{bmatrix}\right).$$

## D.2 Network architectures and hyperparameters

All amortizing policies in our evaluation share a similar structure, with an encoding network that transforms the design-augmented sequences into a stacked representation $\{R(z_s)\}_{s=0}^t$ before passing them into a recurrent layer. Table 2 and Table 3 provide the details.

Moreover, the hyperparameters used for training the policies associated with IO–NPF, IO–SMC$^2$, and iDAD are listed in Table 4 and Table 5. Finally, the evaluation of the learned policies in Table 1 was done with $N = 16$ and $M = 1024$ for the EIG, and with $N = 16$ and $M = 10^5$ for the sPCE.

Table 2: The encoder architecture.

| Layer | Description | Size | Activation |
|---|---|---|---|
| Input | Augmented state $z$ | $\dim(z)$ | - |
| Hidden layer 1 | Dense | 256 | ReLU |
| Hidden layer 2 | Dense | 256 | ReLU |
| Output | Dense | 64 | - |

Table 3: The recurrent network architecture.

| Layer | Description | Size | Activation |
|---|---|---|---|
| Input | $\{R(z_s)\}_{s=0}^t$ | $64 \cdot (t+1)$ | - |
| Hidden layer 1 | GRU | 64 | - |
| Hidden layer 2 | GRU | 64 | - |
| Hidden layer 3 | Dense | 256 | ReLU |
| Hidden layer 4 | Dense | 256 | ReLU |
| Output | Designs $\xi$ | $\dim(\xi)$ | - |

Table 4: Hyperparameters for training the inside–out nested algorithms.

| Hyperparameter | IO–SMC$^2$ (Exact) | IO–SMC$^2$ | IO–NPF (+BS) |
|---|---|---|---|
| N | 32 | 32 | 32 |
| M | – | 128 | 128 |
| Tempering ($\eta$) | 1.0 | 1.0 | 1.0 |
| Slew rate penalty | 0.1 | 0.1 | 0.1 |
| IBIS moves | – | 3 | – |
| Learning rate | $10^{-3}$ | $10^{-3}$ | $10^{-3}$ |
| Training iterations | 25 | 25 | 25 |

Table 5: Hyperparameters for training iDAD.

| Hyperparameter | iDAD |
|---|---|
| Batch size | 512 |
| Number of contrastive samples | 16383 |
| Number of gradient steps | 10000 |
| Learning rate (LR) | $5 \times 10^{-4}$ |
| LR annealing parameter | 0.96 |
| LR annealing frequency (if no improvement) | 400 |

