# OpenReview forum: "Recursive Nested Filtering for Efficient Amortized Bayesian Experimental Design"
_NeurIPS.cc/2024/Workshop/BDU — NeurIPS BDU Workshop 2024 Poster_

### Official Review · Reviewer_ETzF · 2024-09-26

**Rating:** 7
**Confidence:** 2

**Review:**

This paper proposes an algorithm for amortised sequential Bayesian experimental design in a non-exchangeable setting. Overall, the paper is well-organised, and the method is well-motivated and insightful. It builds upon the formulation of Iqbal et al. (2024), but replaces the MCMC kernel with the jittering kernel from the nested particle filter. The authors show that combining the proposed method and the backward sampling scheme improves efficiency while remaining competitive with existing approaches. Overall, I think it is a good work.

---

### Official Review · Reviewer_3uxH · 2024-10-08
**Clear algorithm presentation with good support**

**Rating:** 8
**Confidence:** 3

**Review:**

The reviewer is not deeply familiar with this field, but the authors introduce a new algorithm, IO-NPF, which demonstrates both improved performance and enhanced amortized time efficiency. Overall, the manuscript presents its methodology clearly, provides adequate reference support, and showcases strong empirical results. Therefore, the contributions claimed by the authors are substantiated, making the manuscript a strong candidate for acceptance. However, I have a few comments to share.
1. At line 62, the authors used g_{1:t}(z_{0:t}), which is different from the definition at line 59. It this the multiplication of g at different t?
2. The transition from EIG to equation (1) is too abrupt. It would be better to add some explanation.
3. The empirical results are good, but it would be helpful to include a comparison with more baseline methods. Or including additional experiments with varying datasets or parameters could further validate the algorithm's robustness.

---

### Official Review · Reviewer_RzpJ · 2024-10-09
**Recursive Nested Filtering for Efficient Amortized Bayesian Experimental Design**

**Rating:** 6
**Confidence:** 2

**Review:**

This paper presents a novel recursive algorithm, Inside-Out Nested Particle Filter (IO-NPF), for amortized Bayesian experimental design. The approach is well-motivated, addressing important challenges in sequential design, particularly the computational inefficiencies of existing methods. The introduction of backward sampling to tackle trajectory degeneracy is a noteworthy contribution, and the paper provides both theoretical guarantees and practical validation.

However, the paper's clarity could be improved. Some sections, particularly on backward sampling and the Rao-Blackwellized approach, are dense and could benefit from more detailed explanations or examples to enhance understanding. Additionally, while the experimental results show promise, the scope of the validation is limited to a single example (stochastic pendulum dynamics). Including more diverse case studies would strengthen the generalizability of the proposed method.

The main contribution is incremental but relevant for the field of Bayesian experimental design. The complexity reduction from O(NMT²) to O(NMT) is significant, although the novelty is somewhat limited to the extension of existing particle filtering techniques. Overall, the paper is marginally above the acceptance threshold and would benefit from a more thorough exploration of the method's broader applicability. With revisions, it could make a more substantial impact.

---

### Official Review · Reviewer_1g7P · 2024-10-09
**This paper proposes a novel algorithm, the Inside-Out Nested Particle Filter (IO-NPF), for efficient amortized Bayesian experimental design in sequential settings with non-exchangeable data. The algorithm rephrases policy optimization as a maximum likelihood estimation problem within a non-Markovian state-space model. By leveraging a recursive structure and incorporating a backward sampling scheme to address degeneracy, the IO-NPF aims to improve upon the efficiency and performance of its predecessor, the IO-SMC² algorithm.**

**Rating:** 7
**Confidence:** 4

**Review:**

**Strengths:**

* **Novel Algorithm:** IO-NPF offers a fully recursive, provably consistent alternative to the existing IO-SMC<sup>2</sup>, addressing its non-recursive nature and potential for degeneracy.
* **Theoretical Foundation:** The paper provides a clear theoretical analysis, including convergence guarantees and computational complexity analysis (O(T<sup>2</sup>)) for the IO-NPF.
* **Degeneracy Mitigation:** The authors introduce a Rao-Blackwellized backward sampling technique to effectively counter the problem of trajectory degeneracy, further enhancing the algorithm's performance.
* **Empirical Validation:** Numerical validation on a stochastic pendulum problem demonstrates the superior performance of IO-NPF compared to alternatives, particularly in achieving higher expected information gain (EIG) and the EIG surrogate, Sequential Prior Contrastive Estimate (SPCE).

**Weaknesses:**

* **Limited Experimental Scope:** The numerical validation relies solely on a single example (stochastic pendulum). Evaluating the algorithm on more complex and diverse problems would strengthen the claims and demonstrate its broader applicability.
* **Lack of Comparison:** The paper lacks a comparative analysis with other state-of-the-art amortization methods for Bayesian experimental design, limiting the understanding of IO-NPF's relative advantages and competitiveness.
* **Static Noise Assumption:**  The dual inference formulation assumes static noise in the dynamics. While acknowledged, the paper would benefit from a discussion about the implications and potential limitations if this assumption is violated.
* **Clarity in Appendix:** The explanation of backward sampling and its Rao-Blackwellization in the appendix could be clearer, particularly the notation and derivation of the marginalization step.

**Recommendations:**

* **Expand Experimental Validation:** Evaluate IO-NPF on more challenging and diverse problems to demonstrate its performance and scalability in various settings.
* **Include Comprehensive Comparisons:** Benchmark the IO-NPF against other state-of-the-art amortization methods for Bayesian experimental design, highlighting its relative strengths and weaknesses.
* **Discuss Non-Static Noise Implications:**  Elaborate on the consequences and potential limitations of the static noise assumption and explore possible extensions to handle non-static noise scenarios.
* **Improve Clarity of Appendix:**  Revise the appendix to enhance the clarity of the backward sampling algorithm and its Rao-Blackwellization, ensuring clear notation and a more detailed explanation of the marginalization process.

**Overall Assessment:**

This paper presents a promising new algorithm for amortized Bayesian experimental design. The IO-NPF, with its recursive nature, theoretical guarantees, and improved efficiency, offers a valuable contribution. However, the paper requires minor revisions to address the limited experimental scope, lack of comparisons, and clarity issues.

**Recommendation:**

**Accept with minor revisions.**

The proposed IO-NPF algorithm and its theoretical foundation show strong potential for efficient amortized Bayesian experimental design. Addressing the suggested revisions would solidify the paper's contribution and increase its impact on the field.

---

### Decision · Program_Chairs · 2024-10-09

Accept (Poster)